# Diet, Lifestyle Factors, and Quality of Life in Patients with Rheumatic Diseases: A Cross-Sectional Study

**DOI:** 10.3390/nu17223499

**Published:** 2025-11-07

**Authors:** Gabriela Isabela Răuță Verga, Alexia Anastasia Ștefania Baltă, Silvia Aura Mateescu Costin, Daniela Mihalcia Ailene, Luminița Lăcrămioara Apostol, Tudor Vladimir Gurau, Ciprian Adrian Dinu, Mariana Stuparu-Crețu, Doina Carina Voinescu

**Affiliations:** 1Faculty of Medicine and Pharmacy, University “Dunarea de Jos” of Galati, 800008 Galati, Romania; gabriela.verga@ugal.ro (G.I.R.V.); silviaaura79@gmail.com (S.A.M.C.); daniela.mihalcia@yahoo.com (D.M.A.); gurauvlad16@yahoo.com (T.V.G.); ciprian.dinu@ugal.ro (C.A.D.); doina.voinescu@ugal.ro (D.C.V.); 2Doctoral School of Biomedical Sciences, University “Dunarea de Jos” of Galati, 800008 Galati, Romania; lum.apostol@gmail.com; 3Emergency Clinical Hospital for Children “Sf Ioan”, 800487 Galati, Romania; 4St. “Apostol Andrei” County Emergency Clinical Hospital, 800578 Galati, Romania

**Keywords:** rheumatic diseases, diet, lifestyle, quality of life, nutrition, sleep, physical activity

## Abstract

**Background and Objectives:** Lifestyle and dietary behaviors are increasingly recognized as important factors influencing symptom management and quality of life (QoL) in patients with rheumatic diseases. However, evidence remains limited regarding how specific lifestyle patterns interact with sociodemographic and clinical variables to shape patient-reported outcomes. This study aimed to investigate the relationship between diet, lifestyle behaviors, and self-perceived QoL in a cohort of patients with rheumatic conditions. **Methods:** In this cross-sectional study, 350 adults with rheumatic diseases completed a structured questionnaire covering sociodemographic data, rheumatologic diagnosis and treatment, dietary behaviors, lifestyle factors (physical activity, sleep, smoking, alcohol), and QoL assessments (scales 1–10). Statistical analyses included descriptive measures, Chi-square tests, correlation analyses, logistic regression, and linear regression models to identify predictors of QoL. **Results:** The majority of participants were female (86.9%) and aged between 26 and 55 years. Urban patients were more likely to attribute a positive influence of diet on QoL, while rural participants reported stronger disease burden. Logistic regression showed that adherence to a special diet significantly increased the odds of reporting good QoL. Linear regression identified sleep quality (β = 0.42), perceived dietary influence (β = 0.29), and physical activity (β = 0.18) as independent predictors of QoL (adjusted R^2^ = 0.47, all *p* < 0.001). Correlation analyses further revealed that disease burden negatively impacted emotional well-being and sleep, while dietary influence correlated positively with QoL. **Conclusions:** This study highlights the multidimensional role of diet and lifestyle in shaping QoL in patients with rheumatic diseases. Alongside pharmacological treatment, targeted lifestyle interventions focusing on nutrition, physical activity, and sleep hygiene may substantially improve patient outcomes. Future longitudinal studies are warranted to confirm these associations and explore causal mechanisms.

## 1. Introduction

Rheumatic diseases constitute a heterogeneous group of chronic disorders affecting the musculoskeletal system and, in many cases, multiple organ systems [1]. These conditions, which include rheumatoid arthritis (RA), systemic lupus erythematosus (SLE), psoriatic arthritis (PsA), and various forms of spondyloarthropathies, are characterized by immune-mediated inflammation, progressive joint damage, chronic pain, and functional disability [2,3,4]. Their etiology is multifactorial, involving complex interactions between genetic predisposition, environmental exposures, and dysregulated immune responses [5]. Over the past two decades, research has increasingly emphasized the impact of modifiable lifestyle factors, particularly diet, physical activity, weight management, smoking, alcohol consumption, stress, and sleep, on the development, progression, and management of these conditions [6,7,8,9]. This perspective represents a paradigm shift from a pharmacology-centered approach to a more holistic, biopsychosocial model of rheumatology care [10].

### 1.1. Nutrition as a Modulator of Inflammation

Nutrition is one of the most extensively studied lifestyle factors in rheumatic diseases due to its direct influence on systemic inflammation and immune regulation. Diets high in fruits, vegetables, legumes, whole grains, and unsaturated fats are associated with reduced inflammatory burden and improved functional status in patients with RA and other autoimmune rheumatic conditions [11,12]. Among dietary patterns, the Mediterranean diet has emerged as the most robustly supported [13]. Rich in olive oil, fish, nuts, and polyphenol-containing foods, it has been linked to decreased disease activity, improved quality of life, and even reduced cardiovascular comorbidities, which are a leading cause of mortality in these patients [14,15].

The beneficial effects of the Mediterranean diet are mediated partly through the intake of omega-3 polyunsaturated fatty acids (PUFAs), which reduce the production of pro-inflammatory eicosanoids and cytokines [16]. Several randomized controlled trials have demonstrated that omega-3 supplementation decreases joint pain, morning stiffness, and the need for non-steroidal anti-inflammatory drugs (NSAIDs) in RA patients [17]. Similarly, antioxidants such as vitamins C and E, selenium, and polyphenols exert protective effects by counteracting oxidative stress, which plays a pivotal role in joint damage and disease progression [18].

On the other hand, Western dietary patterns characterized by high consumption of red and processed meats, refined sugars, saturated fats, and ultra-processed foods are consistently associated with increased systemic inflammation and poorer disease outcomes [19]. These diets contribute to obesity, insulin resistance, and metabolic syndrome, all of which aggravate inflammatory responses and accelerate joint damage [20]. Thus, the evidence strongly supports dietary modification as an integral component of rheumatic disease management.

### 1.2. Physical Activity and Weight Management

While nutrition provides a biochemical foundation for immune modulation, physical activity represents another critical pillar of lifestyle intervention [21]. Historically, patients with rheumatic diseases were often advised to avoid exercise due to concerns about exacerbating joint damage [22]. However, contemporary evidence indicates that appropriately tailored physical activity confers substantial benefits without increasing structural harm [23]. Regular exercise improves joint mobility, preserves muscle mass, enhances cardiovascular health, and alleviates fatigue and depressive symptoms [24].

Low-impact aerobic exercises such as swimming, cycling, and walking, combined with strength training and flexibility routines, have shown particular promise [25]. Yoga and tai chi, for example, not only improve physical function but also reduce stress and enhance psychological well-being [26]. Importantly, exercise interventions need to be individualized, considering the patient’s disease activity, comorbidities, and functional limitations [27].

Weight management is also central to disease outcomes. Excess adiposity not only increases mechanical load on joints but also acts as a source of pro-inflammatory adipokines, such as leptin and resistin, which perpetuate systemic inflammation [28,29]. Obesity has been associated with higher disease activity scores, poorer treatment response to biologic therapies, and increased risk of comorbidities such as cardiovascular disease and type 2 diabetes [30,31]. Accordingly, weight reduction through combined dietary and physical activity strategies can yield both mechanical and immunological benefits [32].

### 1.3. Smoking, Alcohol, and Other Lifestyle Factors

Beyond diet and exercise, other lifestyle factors significantly influence disease progression. Smoking is one of the most well-established environmental risk factors for RA [33]. It increases the risk of disease onset, particularly in genetically predisposed individuals carrying the HLA-DRB1 shared epitope, and it worsens disease severity by amplifying autoantibody production, including anti-citrullinated protein antibodies (ACPA) [34]. Furthermore, smokers exhibit reduced therapeutic response to disease-modifying antirheumatic drugs (DMARDs) and biologics, leading to poorer long-term outcomes [35].

Alcohol consumption presents a more nuanced picture. Moderate intake has been associated in some studies with reduced RA risk, possibly due to its anti-inflammatory and immunomodulatory effects [36]. However, excessive alcohol use increases systemic inflammation, worsens comorbidities, and can cause hepatotoxicity, especially when combined with methotrexate or other hepatotoxic drugs [37]. As such, clinical recommendations emphasize moderation and individualized assessment.

### 1.4. Psychological Health and Sleep Quality

The psychosocial dimension of lifestyle is equally critical in rheumatic diseases. Stress, anxiety, and depression are highly prevalent among these patients and contribute to higher pain perception, fatigue, and disability [38]. Stress-induced activation of the hypothalamic–pituitary–adrenal (HPA) axis and sympathetic nervous system can exacerbate immune dysregulation, creating a vicious cycle of inflammation and psychological distress [39].

Sleep quality represents another underrecognized determinant of disease outcomes. Poor sleep, often resulting from nocturnal pain or comorbid mood disorders, leads to heightened fatigue, impaired immune function, and amplified inflammatory activity [40]. Cognitive–behavioral therapy for insomnia, relaxation techniques, and mindfulness-based interventions have shown efficacy in improving sleep quality and overall well-being in this population [41,42].

### 1.5. Toward Integrative Management

Taken together, the available evidence underscores that rheumatic disease management cannot rely solely on pharmacological therapy. While disease-modifying drugs remain essential for controlling inflammation and preventing structural damage, lifestyle interventions significantly enhance therapeutic outcomes [43]. Nutritional counseling, regular physical activity, weight control, smoking cessation, moderation of alcohol intake, and psychological support should be incorporated into comprehensive care strategies [44,45,46].

## 2. Materials and Methods

### 2.1. Design

This study was conducted as a cross-sectional, non-interventional survey designed to evaluate dietary habits and lifestyle behaviors among patients with rheumatic diseases. Data were collected through a structured, self-administered questionnaire (Appendix A), distributed during routine clinical visits. The study design was chosen to capture a wide range of lifestyle-related variables without interfering with patients’ ongoing medical management.

Data were collected through a structured, self-administered questionnaire developed by the research team.

Patients were enrolled from the Emergency Clinical Hospital for Children “Sf. Ioan”, and the St. “Apostol Andrei” County Emergency Clinical Hospital, Galați, Romania. The study design was chosen to capture a wide range of lifestyle-related variables without interfering with patients’ ongoing medical management.

Participation was voluntary and anonymous, and informed consent was obtained prior to questionnaire completion. Because recruitment relied partly on regional hospital participation, the study may be subject to selection bias, favoring participants with access to specialized care within the Galați region.

### 2.2. Participants

All adult patients (≥18 years) with a confirmed diagnosis of rheumatic disease, including rheumatoid arthritis, systemic lupus erythematosus, psoriatic arthritis, or other connective tissue disorders, were eligible to participate. A total of 350 questionnaires were fully completed and included in the analysis. Exclusion criteria encompassed severe cognitive impairment, inability to understand the questionnaire, or refusal to provide informed consent. Recruitment was consecutive, and all patients were approached during the predefined study period to achieve the target sample size.

Patients were stratified into subgroups according to age, sex, body mass index (BMI), and smoking status to allow comparative analyses of lifestyle factors across demographic and clinical categories.

### 2.3. Data Collection

Data collection was carried out between July–September by trained research staff. Participants completed a structured questionnaire during outpatient visits, with assistance from healthcare professionals when needed to ensure completeness and accuracy.

The questionnaire included sections on sociodemographic data, rheumatologic diagnosis and treatment, dietary habits, lifestyle behaviors (physical activity, sleep, alcohol and tobacco use), and self-perceived quality of life.

In this study, the term “special diet” refers to any dietary pattern that deviates from a regular, unrestricted diet and is intentionally adopted to manage health, control body weight, or alleviate disease-related symptoms. Such diets may be medically prescribed by a healthcare professional (e.g., low-sodium, low-fat, hypocaloric, or gluten-free diets) or self-initiated by patients (e.g., Mediterranean, vegetarian, or lactose-free diets).

The questionnaire collected the following:

Sociodemographic data: age, gender, marital status, education, and employment.

Clinical data: type and duration of rheumatic disease, treatment regimens, comorbidities.

Lifestyle variables: smoking status, alcohol consumption, sleep quality, stress levels, and physical activity.

Dietary variables: frequency of fruit, vegetable, fish, whole grain, and processed food consumption, adherence to the Mediterranean diet, use of supplements (e.g., omega-3, vitamins).

Completion time was approximately 15–20 min. Only patients who were conscious, communicative, and capable of independent self-reporting were included.

### 2.4. Instrument

The questionnaire was developed by the research team based on validated instruments used in nutrition and lifestyle research, adapted for patients with rheumatic conditions. It incorporated both closed-ended and Likert-scale items to assess dietary frequency, lifestyle behaviors, and self-perceived health.

Dietary assessment was based on a semi-quantitative food frequency questionnaire (FFQ).

Lifestyle assessment included the International Physical Activity Questionnaire (IPAQ) short form and items adapted from the Pittsburgh Sleep Quality Index (PSQI).

A self-rated health scale (0 = very poor, 10 = excellent) was used to capture patients’ overall perception of well-being.

The questionnaire was pilot-tested on a small subsample (n = 20) to ensure clarity and feasibility prior to full administration.

### 2.5. Statistics

Data were analyzed using IBM SPSS Statistics for Windows, Version 26.0 (IBM Corp., Armonk, NY, USA). Descriptive statistics (mean, standard deviation, median, interquartile range, and frequency distributions) were used to summarize sociodemographic, dietary, and lifestyle characteristics.

Inferential analyses included the following:χ^2^ tests for associations between categorical variables.*t*-tests for group comparisons based on continuous variables.Logistic regression to explore predictors of unhealthy lifestyle patterns.Statistical significance was set at *p* < 0.05. Missing data were excluded case-wise.

### 2.6. Ethics

The study was approved by the Ethics Committee of the “Dunărea de Jos” University of Galați in May 2023. All participants provided informed consent after receiving both written and verbal explanations of the study objectives, procedures, and data protection protocols. Participation was voluntary, and patients were informed that they could withdraw at any point without any effect on their clinical care.

## 3. Results

### 3.1. Sociodemographic Characteristics

A total of 350 patients with rheumatic diseases completed the questionnaire. The majority of participants were aged between 26 and 55 years, with a predominance of females (86.9%) and most residing in urban areas. In terms of education, post-secondary and higher education levels were most frequently reported, while only a minority had completed primary school. The largest occupational group was represented by employed individuals, followed by retirees, with few reporting unemployment. Only 12% of patients were classified with a disability due to their rheumatic condition.

Regarding economic and lifestyle characteristics, more than half of the respondents reported spending 26–50% of their monthly income on food. Most participants had an average height between 150 and 170 cm and a body weight within the 60–80 kg range, while extreme values for height and weight were uncommon (Table 1).

### 3.2. Rheumatologic Diagnoses and Treatments

Table 2 presents the distribution of rheumatologic diagnoses, treatment modalities, and related characteristics among the 350 participants. Osteoarthritis (41%) and rheumatoid arthritis (32.8%) were the most prevalent conditions, followed by ankylosing spondylitis (16.9%) and systemic lupus erythematosus (8.2%). Less common diseases included gout (6.6%), vasculitis (5.5%), Sjögren’s syndrome (3.8%), polymyalgia rheumatica (2.2%), and scleroderma (1.6%).

Regarding disease duration, most patients had been diagnosed for 6–11 years (≈46.5%), while only 23.5% reported a disease history shorter than 1 year. Despite the chronicity of illness, only 43.7% were regularly followed-up by a rheumatologist, highlighting a potential gap in specialist care.

The most frequently reported treatments were supplements (64.5%), NSAIDs (56.3%), and analgesics (37.7%). On the other hand, corticosteroids (13.7%), DMARDs (15.8%), and biologics (6.6%) were used less commonly. Treatment duration varied, with one-third of patients undergoing therapy for less than six months, and another third for over three years, indicating a heterogeneous pattern of long- versus short-term management.

Complementary therapies were also reported, with physiotherapy (56.9%) being most common, followed by homeopathy (19.1%) and acupuncture (16.9%). Surgical interventions related to rheumatic disease were rare (7.7%).

When asked about the perceived effect of treatment on body weight, responses were mixed, though the largest subgroup (27.9%) considered the impact to be very low, while 26.2% reported moderate-to-high influence. Adherence to therapy was suboptimal, with only 57.4% reporting regular use as prescribed, and 42.6% indicating irregular or occasional intake.

The inclusion of multiple rheumatic diseases in this study was intentional and based on the objective of exploring general lifestyle and dietary patterns that may influence quality of life across the broader rheumatologic population. Although these diseases differ in pathophysiology, inflammatory activity, and clinical manifestations, they share common features such as chronic pain, fatigue, physical limitations, and the need for long-term treatment and self-management.

By including a heterogeneous sample, the study aimed to capture shared lifestyle-related challenges and behaviors among patients living with chronic rheumatic conditions, rather than focusing on disease-specific outcomes. This approach allows for a more comprehensive understanding of how diet, sleep, and physical activity affect overall well-being in this patient population. However, this choice may reduce the specificity of the diagnosis, an issue that should be addressed in future studies through subgroup analyses or disease-specific comparisons.

### 3.3. Dietary Behaviors and Lifestyle

Table 3 presents dietary habits and lifestyle behaviors among the study participants. Food intolerances were relatively uncommon (18.6%), while the majority reported no restrictions. Most patients consumed 2–3 meals daily (80.5%), with a smaller proportion following a structured pattern of three meals and snacks (18.3%).

Unhealthy dietary behaviors such as frequent fast-food consumption were rare, with nearly 60% reporting rare or no intake, and only 15% consuming fast food weekly or more. On the other hand, home-cooked meals were dominant (88%), underscoring a preference for traditional eating habits.

When analyzing protein sources, chicken was consumed most frequently, with over half reporting intake at least twice per week. Beef consumption was markedly lower, with more than half eating it monthly and nearly 30% rarely or never. Fish consumption was moderate, with about half of participants consuming it weekly, while seafood intake was generally less frequent, with nearly 40% reporting rare or no consumption.

Dairy preferences varied: whole cow’s milk (25.1%) and plant-based alternatives (15.7%) were common, but a significant proportion (37.9%) reported not consuming milk at all. Regarding cooking fats, vegetable oils (66.9%) were most frequently used, followed by olive oil (16.3%), while animal fats such as lard were reported by only 6.6%.

Water intake was adequate in most cases, with over 80% reporting consumption of at least 1–3 L daily, and very few drinking less than 1 L. Cooking methods reflected a preference for healthier practices such as baking (59.7%) and boiling (58.6%), though less healthy practices like deep frying were reported by only 1.7%.

Overall, these findings suggest that participants with rheumatic diseases generally adhere to balanced dietary patterns, with a preference for home-cooked meals, moderate fish intake, and healthier cooking methods, while the prevalence of unhealthy practices such as fast-food and deep-fried consumption was relatively low.

### 3.4. Perceptions of Diet and Its Impact on Symptoms Among Study Participants

Table 4 illustrates participants’ perceptions regarding diet and its impact on disease symptoms. Nearly half of respondents (48.3%) reported qualitative changes in their diet after diagnosis, while a smaller group (8.7%) modified their diet quantitatively, and 43% made no changes.

When asked about beneficial foods, vegetables (85.1%) and fruits (80.0%) were most frequently recognized as supportive for health, followed closely by healthy fats (83.1%), legumes (79.7%), lean proteins (76.9%), and nuts/seeds (76.6%). On the other hand, dairy products were less consistently perceived as beneficial (52.3%).

Foods most often reported as aggravating symptoms included sweets and sugary products (58.6%), soft drinks (58.6%), alcohol (51.4%), and refined processed foods (50.6%). Gluten-containing foods, spicy meals, and red/processed meat were also frequently implicated as triggers.

Regarding symptom control, only a minority perceived diet as having a very high (12.0%) or high (18.6%) effect on pain reduction, whereas most rated its influence as low to moderate. Similarly, while 22% believed diet strongly influences disease symptoms, nearly one-third considered the effect minimal.

Lifestyle habits revealed that alcohol use was limited, with almost half (44.6%) reporting abstinence, and only 2.8% drinking often. Smoking prevalence was low, as 80.9% of participants reported never smoking. Finally, 21.7% followed a specific diet, while the majority (78.3%) did not.

Overall, these findings suggest that patients with rheumatic diseases are aware of the role of diet, particularly the benefits of plant-based foods and healthy fats, while recognizing potential triggers such as processed and sugary foods. However, the perceived direct impact of diet on symptom improvement remains variable, indicating a need for clearer dietary guidance in clinical practice.

### 3.5. Self-Perceived Quality of Life and Lifestyle Perceptions

Table 5 presents patients’ self-assessment of quality of life (QoL) and lifestyle-related perceptions. When rating the impact of rheumatic disease on their lives, most respondents scored between 3 and 6, indicating moderate to substantial impairment. However, current overall QoL scores tended to be higher, with 43.7% rating their present QoL as good (7–8) and 16.9% as very good (9–10).

Disease control was perceived positively by 52.6%, though almost one-third reported only partial control. The disease had a notable negative impact on social relationships, with nearly half scoring 1–4, while emotional well-being was also affected, most respondents placing themselves in the low-to-moderate range.

Sleep quality was frequently compromised, with almost 20% reporting strong disturbances and one-third being neutral. Daily activities were moderately affected, as nearly half of participants indicated a medium-to-high burden.

In terms of physical activity, walking was the most frequent form (61.1% reporting high engagement), whereas running and structured sports were less common. Exercises performed at home or in therapy programs were reported at moderate levels (48%).

Regarding dietary recommendations, participants most frequently endorsed the Mediterranean diet (47.4%) and high-fiber diet (45.1%), followed by hypocaloric, low-carb, and low-fat regimens. Gluten-free, lactose-free, and vegetarian/vegan diets were much less popular.

Finally, when asked to what extent diet influences QoL, nearly two-thirds (65.1%) rated this effect as high or very high (scores 7–10), underscoring a strong patient belief in the role of nutrition as a supportive therapy in rheumatic disease.

### 3.6. Self-Perceived Quality of Life and Dietary Influence in Patients with Rheumatic Diseases

Figure 1 illustrates the distribution of patient responses on a scale from 1 to 10, where 1 represents a very poor perception and 10 a very good perception. Three dimensions were assessed: the extent to which nutrition influences quality of life, the degree to which rheumatic disease affects quality of life, and the overall appreciation of current quality of life.

The results presented in Figure 1 highlight notable differences between the three assessed domains.

Nutrition and quality of life: Most patients rated the influence of diet on their well-being in the moderate-to-high range (scores 5–8), with peaks at 5 (19.1%) and 8 (17.4%). A smaller group considered diet to have very little impact (scores 1–2 = 6.3%), while 9.1% rated the impact as maximal (score 10). This suggests that dietary habits are widely perceived as relevant but not universally decisive in shaping quality of life.

Rheumatic disease and quality of life: The disease was perceived as having a consistently negative impact, with higher frequencies at the lower-to-moderate scores (3–6 = 57.4%). Very positive perceptions were rare (scores 9–10 = 11.4%). This pattern reflects the significant burden of rheumatic conditions on daily functioning, though some patients maintain a more resilient outlook.

Overall appreciation of quality of life: On the other hand, current quality of life was assessed more favorably, with the largest proportion of responses clustering at scores 7–8 (47.7%). Very low scores (1–2) were uncommon (1.7%), indicating that despite disease-related challenges, most participants sustain a satisfactory perception of life quality.

Taken together, these findings suggest that while rheumatic disease significantly burdens patients, nutrition is perceived as a supportive factor that can mitigate its impact. Moreover, the relatively high self-reported quality of life indicates adaptive coping strategies and possibly effective management of the disease.

### 3.7. Comparison of Self-Perceived Quality of Life Scores by Sociodemographic Groups

Table 6 presents the comparison of self-reported quality of life dimensions according to key sociodemographic characteristics, including place of residence (urban vs. rural), gender, and age (<55 vs. ≥55 years). The outcomes considered were the perceived impact of rheumatic disease on quality of life (Q39), current self-assessed quality of life (Q40), and the perceived influence of nutrition on quality of life (Q48).

The findings indicate several statistically significant differences across sociodemographic groups. Patients residing in rural areas reported a slightly greater impact of disease on their quality of life compared with their urban counterparts (*p* = 0.041). Gender-based analysis revealed that women perceived a stronger negative impact of the disease (mean score 5.4) compared to men (mean score 6.1, *p* = 0.030), reflecting a potentially higher disease burden among female patients.

Age also emerged as a relevant factor. Participants aged ≥55 years perceived a greater impact of rheumatic disease on their daily life (mean score 6.0) compared with those below 55 years (mean score 5.2, *p* = 0.012). Similarly, younger patients reported significantly higher scores for current quality of life (*p* = 0.045), suggesting that age-related comorbidities may further reduce quality of life in older adults.

With regard to the role of nutrition, urban residents tended to attribute a stronger positive influence of diet on quality of life compared with rural respondents (mean scores 7.1 vs. 6.5, *p* = 0.032). No significant gender differences were observed for this variable, though a trend towards greater recognition of dietary influence was noted among younger participants.

Overall, these results underscore the importance of considering sociodemographic factors when evaluating patients’ self-perceived quality of life, particularly the differential burden of disease across gender and age, and the potential role of nutrition as a modifiable factor more strongly acknowledged in urban populations.

### 3.8. Correlation Matrix of Dietary Influence, Disease Impact, Emotional State, Sleep Quality, and Quality of Life

Table 7 presents the correlation coefficients between self-reported scores of disease impact, current quality of life, emotional state, sleep quality, and the perceived influence of diet. The analysis was conducted to explore interrelationships among these key dimensions of patient-reported outcomes.

As shown in Table 7, a moderate positive correlation was observed between the perceived influence of diet (Q48) and current quality of life (Q40) (r = 0.36, *p* < 0.001). On the other hand, disease impact (Q39) was strongly and negatively correlated with both current QoL (r = −0.52, *p* < 0.001) and emotional well-being (r = −0.48, *p* < 0.001), and moderately with sleep quality (r = −0.41, *p* < 0.001). These results suggest that greater disease burden is associated with poorer emotional status and sleep, whereas positive perceptions of diet are linked to better quality of life outcomes.

### 3.9. Associations Between Dietary Behaviors and Patient Perceptions

Table 8 presents the results of Chi-square tests examining associations between dietary behaviors and patient perceptions, specifically between adherence to a rheumatic disease-adapted diet and perceived pain improvement, as well as between education level and fast-food consumption frequency. Patients who followed a diet adapted to rheumatic disease were significantly more likely to report a large or very large improvement in pain through nutrition compared to those who did not follow such a diet (55.3% vs. 23.7%, *p* < 0.001).

There was also a significant association between education level and fast-food consumption: participants with higher education reported significantly lower frequencies of fast-food consumption compared to those with primary or secondary education (*p* = 0.007). This observation, although secondary to the main objectives of the study, provides relevant information on factors that may influence eating behaviors and, implicitly, the quality of life of patients. The level of education could reflect a higher degree of awareness of the effects of nutrition on health and an increased ability to access appropriate food information and resources.

Table 9 presents the results of Chi-square tests examining associations between dietary behaviors and patient perceptions, specifically between adherence to a special diet and perceived pain improvement, and between education level and frequency of fast-food consumption.

Patients following a special diet were significantly more likely to report high or very high improvement in pain through nutrition compared with those not following a diet (55.3% vs. 23.7%, *p* < 0.001). Similarly, education was strongly associated with fast-food consumption: participants with higher education reported significantly lower frequencies of fast-food intake compared with those with primary or secondary education (*p* = 0.007).

### 3.10. Multivariate Analyses

Table 10 presents the results of a linear regression model using current self-reported quality of life (Q40) as the dependent variable, with sleep quality (Q44), perceived dietary influence (Q48), and regular physical activity (Q46) as predictors.

The model explained nearly half of the variance in quality of life scores (Adjusted R^2^ = 0.47). All three predictors were statistically significant. Sleep quality emerged as the strongest predictor (β = 0.42, *p* < 0.001), followed by perceived dietary influence (β = 0.29, *p* < 0.001) and regular physical activity (β = 0.18, *p* < 0.001).

These findings indicate that better sleep, a stronger belief in the role of diet, and engagement in physical activity are independently associated with higher quality of life in patients with rheumatic diseases. Lifestyle and behavioral factors thus account for a substantial proportion of the variability in patients’ well-being.

### 3.11. Visual Analyses

Table 11 illustrates the correlation matrix between quality of life, social relations, emotional state, sleep quality, physical activity, and dietary influence.

The heatmap shows **moderate positive correlations** among most dimensions, with the strongest associations observed between emotional state and social relations (r = 0.52) and between current quality of life and emotional state (r = 0.45). Sleep quality also correlated positively with both emotional well-being and QoL, while dietary influence, although weaker, still showed consistent positive links. This pattern suggests that multiple lifestyle and psychosocial factors interact to shape patients’ perception of quality of life.

Figure 2 shows a radar chart comparing urban and rural patients across six dimensions of quality of life: social relations, emotional state, sleep quality, daily activities, physical activity, and dietary influence.

The chart demonstrates that urban patients scored consistently higher across all dimensions, with the largest gaps observed in dietary influence (7.2 vs. 6.3) and daily activities (6.9 vs. 6.0). Rural patients reported lower scores in sleep and emotional well-being, indicating a more vulnerable profile. This visualization underscores disparities by place of residence and highlights areas where rural patients may require targeted support.

## 4. Discussion

The findings of this study provide new insights into the complex relationship between diet, lifestyle behaviors, and perceived quality of life in patients with rheumatic diseases. The analysis revealed that a substantial proportion of patients recognized the role of diet in modulating symptoms and quality of life, with higher dietary awareness being associated with improved self-reported well-being (Q48 vs. Q40). These results suggest that nutritional factors are not only relevant for metabolic health but may also influence the subjective disease experience, aligning with recent evidence supporting the anti-inflammatory and symptomatic benefits of Mediterranean and anti-inflammatory diets in rheumatology [2,6,11,47].

A second important observation was the strong negative association between disease burden (Q39) and emotional well-being and sleep quality (Q43–Q44). Patients who perceived the disease as having a high impact also reported greater emotional distress and more sleep disturbances. This aligns with existing literature highlighting the bidirectional relationship between chronic pain, fatigue, and psychological health in rheumatic conditions [3,4,5,46]. In particular, studies in rheumatoid arthritis and lupus populations have emphasized how pain and inflammation disrupt circadian rhythm and sleep, which in turn exacerbates fatigue and reduces coping capacity [8,11,21].

Sociodemographic comparisons further contextualized these findings. Urban patients reported higher scores for the beneficial role of diet, potentially reflecting greater access to nutritional information and healthier food options, whereas rural participants tended to perceive a stronger negative impact of the disease. Gender differences were also notable: female patients reported lower quality of life and greater symptom burden, consistent with prior evidence that women are disproportionately affected by both the prevalence and severity of rheumatic diseases [19,25,31]. Age stratification showed that younger patients (<55 years) tended to report better overall quality of life, while older patients experienced greater disease burden and poorer sleep, underlining the cumulative effect of disease progression and comorbidities.

The regression models offered additional insights. Logistic regression indicated that adherence to a special diet significantly increased the odds of reporting good quality of life, even after adjusting for sociodemographic and treatment-related variables. Similarly, linear regression identified sleep quality, dietary influence, and physical activity as independent predictors of current quality of life, with sleep emerging as the strongest factor. These results emphasize the importance of lifestyle interventions alongside pharmacological management, echoing calls in the literature for holistic approaches in rheumatology care [10,23,42].

Visual analyses strengthened these findings. Heatmaps highlighted consistent positive correlations between emotional well-being, social relations, sleep, and quality of life, with diet exerting a moderate but meaningful contribution. Cluster analysis identified distinct patient profiles, such as younger urban individuals with healthier habits and higher QoL versus older, sedentary patients with poorer outcomes. Radar charts underscored disparities between urban and rural groups, with urban patients reporting advantages across most lifestyle dimensions. Such visualizations provide a nuanced understanding of patient subgroups and can guide tailored interventions.

Despite these important insights, several limitations should be acknowledged. First, the cross-sectional design precludes causal inference, limiting the ability to determine whether dietary changes directly improve quality of life or are simply associated with greater health awareness. Second, the reliance on self-reported questionnaires may introduce reporting bias, particularly in lifestyle behaviors such as diet and physical activity. Third, while the sample size was relatively large (N = 350), stratified subgroup analyses (e.g., by disease type or treatment modality) may have been underpowered. Moreover, dietary assessment relied on simplified frequency questions rather than detailed dietary recalls, which restricts the ability to quantify nutrient intake or adherence to specific dietary patterns.

Another limitation of the study is the possibility of reporting errors due to individual perceptions, beliefs about nutrition, and the accuracy of patients’ memories. Since the diet assessment was based on self-reporting, the results may reflect patients’ perceptions of their own eating behavior rather than an objective measurement of it. Furthermore, as a cross-sectional study, it allows for the identification of associations but cannot demonstrate causal relationships between eating behaviors and quality of life. Additional prospective studies are needed to confirm these results.

Future research should build on these results through longitudinal and interventional studies. Prospective designs could clarify whether sustained dietary modifications or structured exercise programs lead to measurable improvements in symptoms, disease activity, and quality of life. Additionally, integrating biomarkers of inflammation and metabolic health would allow exploration of the biological mechanisms linking lifestyle factors with disease outcomes. Finally, intervention studies tailored to vulnerable subgroups, such as women and rural populations, could help to reduce disparities and promote equity in rheumatology care.

The statistical analyses were exploratory in nature, aiming to identify possible associations between eating behaviors, patient perceptions, and quality of life. *p*-values were interpreted as descriptive indicators of the strength of the observed relationships, without applying a correction for multiple testing. This approach is in line with similar exploratory studies in the literature, but the results should be interpreted with caution, considering the possibility of statistically significant associations due to repeated testing. Consequently, further research with a prospective design and larger samples is needed to confirm these findings.

Overall, this study highlights that while rheumatic diseases significantly impair quality of life, lifestyle behaviors particularly nutrition, sleep, and physical activity can mitigate their impact. Incorporating lifestyle counseling into standard rheumatology practice may offer a low-cost and patient-centered strategy to complement pharmacological treatment and improve long-term outcomes.

## 5. Conclusions

In this study of patients with rheumatic diseases, diet and lifestyle factors were shown to play an important role in shaping self-perceived quality of life. Patients who reported adhering to a special diet and engaging in regular physical activity were more likely to rate their quality of life positively, while poor sleep quality and higher perceived disease burden were strongly associated with unfavorable outcomes. Significant differences were observed across sociodemographic groups, with women and older patients reporting greater impairment, and urban patients demonstrating higher awareness of dietary influence. These findings provide evidence that, alongside pharmacological treatment, lifestyle interventions particularly promoting healthy dietary habits, physical activity, and sleep hygiene are essential in the comprehensive management of rheumatic diseases.

## Figures and Tables

**Figure 1 nutrients-17-03499-f001:**
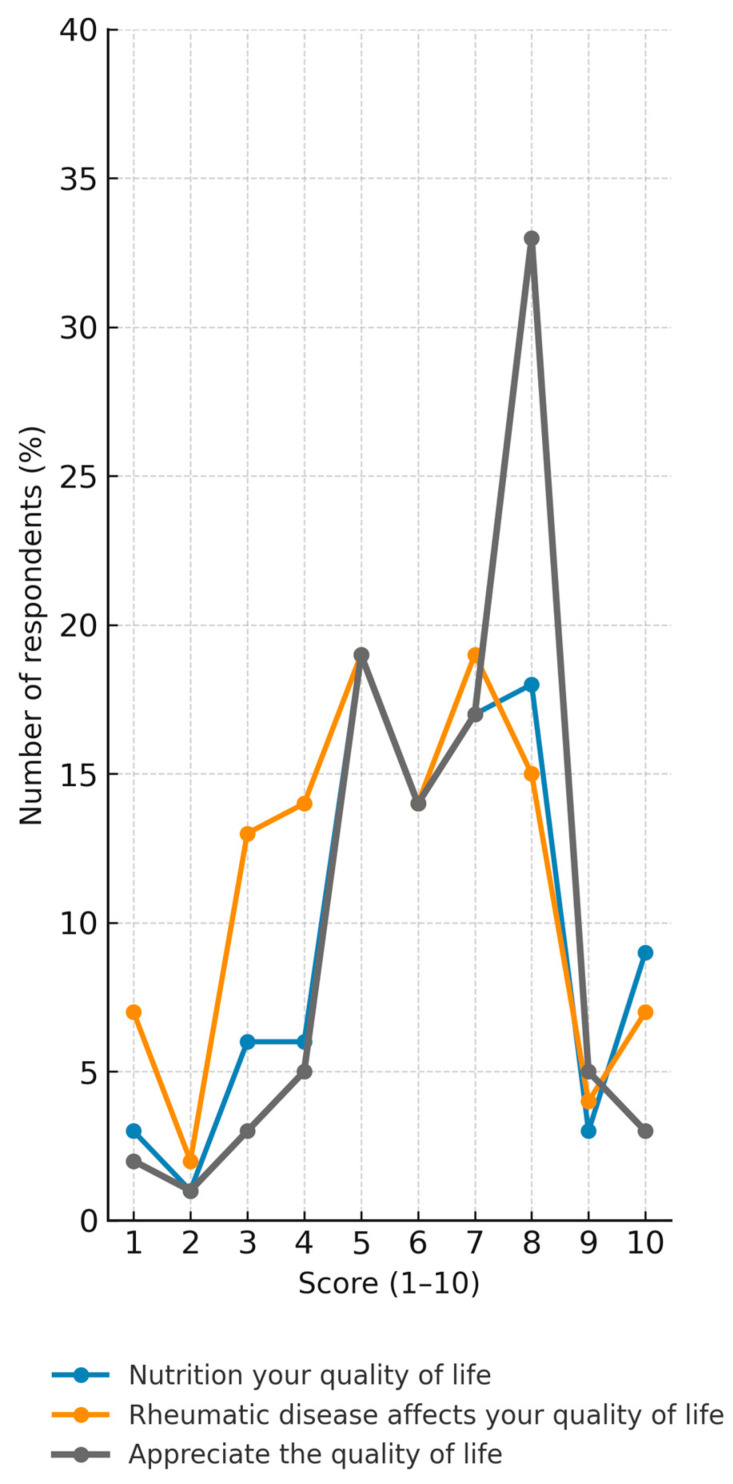
Self-perceived quality of life and dietary influence in patients with rheumatic diseases.

**Figure 2 nutrients-17-03499-f002:**
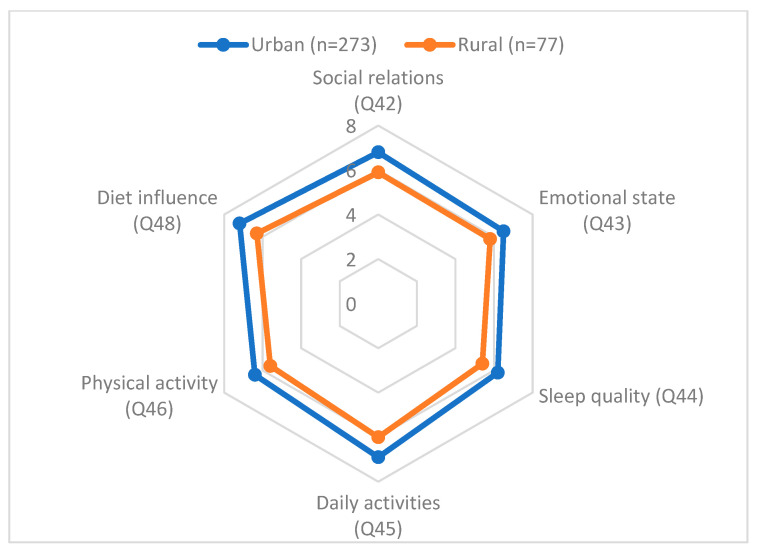
Radar chart comparing urban and rural patients across six dimensions of quality of life.

**Table 1 nutrients-17-03499-t001:** Sociodemographic and economic characteristics of study participants (N = 350).

Variable	Category	N	%
**Age (years)**	18–25	10	2.7
	26–35	27	7.7
	36–45	67	19.1
	46–55	124	35.5
	>55	122	35.0
**Gender**	Male	46	13.1
	Female	304	86.9
**Residence**	Urban	273	78.1
	Rural	77	21.9
**Education**	Primary	21	6.0
	Secondary	67	19.1
	Post-secondary	113	32.2
	Bachelor’s degree	56	16.0
	Master’s degree	84	24.0
	Doctorate	9	2.7
**Occupation**	Unemployed	1	0.3
	Housewife	21	6.0
	Student	11	3.0
	Freelancer	5	1.5
	Employee	228	65.0
	Manager	7	2.0
	Retired	77	21.9
**Disability status**	Yes	42	12.0
	No	308	88.0
**Income spent on food**	10–25%	71	20.2
	26–50%	189	54.1
	51–70%	77	21.9
	>70%	13	3.8
**Height (cm)**	140–150	8	2.2
	150–160	92	26.2
	160–170	186	53.0
	170–180	54	15.3
	180–190	9	2.7
	>190	1	0.3

**Table 2 nutrients-17-03499-t002:** Rheumatologic diagnoses and treatments among study participants (N = 350).

Variable	Category	N	%
Diagnosis	Rheumatoid arthritis	115	32.8
	Ankylosing spondylitis	59	16.9
	Systemic lupus erythematosus	29	8.2
	Sjögren’s syndrome	13	3.8
	Vasculitis	19	5.5
	Osteoarthritis	144	41.0
	Gout	23	6.6
	Polymyalgia rheumatica	8	2.2
	Scleroderma	5	1.6
Disease duration	<1 year	82	23.5
	1–2 years	27	7.7
	3–5 years	33	9.3
	6–8 years	94	26.8
	9–11 years	69	19.7
	12–15 years	38	11.0
	16–20 years	14	4.0
	>20 years	49	14.0
Follow-up with rheumatologist	Yes	153	43.7
	No	197	56.3
Current treatment	NSAIDs	197	56.3
	Corticosteroids	48	13.7
	DMARDs	55	15.8
	Biologics	23	6.6
	Analgesics	132	37.7
	Supplements (vitamin D, calcium)	226	64.5
Treatment duration	<6 months	117	33.3
	6–12 months	59	16.9
	1–3 years	61	17.5
	>3 years	113	32.3
Complementary therapies	Physiotherapy	199	56.9
	Acupuncture	59	16.9
	Homeopathy	67	19.1
Surgery due to rheumatic disease	Yes	27	7.7
	No	323	92.3
Perceived effect of treatment on body weight	Very high	33	9.3
	High	59	16.9
	Neutral	30	8.7
	Low	63	18.0
	Very low	98	27.9
	None	67	19.1
Treatment adherence	Regular	201	57.4
	Irregular	40	11.5
	Occasional	109	31.1

**Table 3 nutrients-17-03499-t003:** Dietary behaviors and lifestyle characteristics of study participants (N = 350).

Variable	Category	N	%
**Food intolerances**	Yes	65	18.6
	No	285	81.4
**Daily meals**	1 meal	4	1.1
	2 meals	134	38.3
	3 meals	148	42.3
	3 meals + snacks	64	18.3
**Fast-food consumption**	Daily	0	0.0
	Twice/week	27	7.7
	Weekly	25	7.1
	Monthly	90	25.7
	Rare/Never	208	59.5
**Home-cooked meals**	Daily	308	88.0
	Twice/week	29	8.3
	Weekly	13	3.7
**Chicken meat**	Daily	44	12.6
	Twice/week	135	38.6
	Weekly	134	38.3
	Monthly	32	9.1
	Rare/Never	5	1.4
**Beef meat**	Daily	2	0.6
	Twice/week	17	4.9
	Weekly	44	12.6
	Monthly	184	52.5
	Rare/Never	103	29.4
**Fish consumption**	Daily	4	1.1
	Twice/week	52	14.9
	Weekly	172	49.1
	Monthly	78	22.3
	Rare/Never	44	12.6
**Seafood consumption**	Daily	2	0.6
	Twice/week	15	4.3
	Weekly	124	35.4
	Monthly	73	20.9
	Rare/Never	136	38.8
**Type of milk**	Whole cow’s milk	88	25.1
	Skimmed/semi-skimmed	33	9.3
	Lactose-free	28	8.0
	Plant-based (soy, almond, oat, etc.)	55	15.7
	Goat milk	14	4.0
	None	132	37.9
**Cooking fat**	Vegetable oil	234	66.9
	Olive oil	57	16.3
	Butter	15	4.3
	Lard/tallow	23	6.6
	Coconut/palm oil	2	0.6
	Other oils	16	4.6
	Do not cook	3	0.9
**Water intake**	0–1 L	65	18.6
	1–2 L	126	36.0
	2–3 L	154	44.0
	3–4 L	5	1.4
	None/Don’t know	0	0.0
**Preferred cooking methods**	Boiling	205	58.6
	Baking (oven)	209	59.7
	Grilling	142	40.6
	Air fryer	81	23.1
	Pan frying	84	24.0
	Deep frying	6	1.7
	Sautéing	40	11.4
	Steaming	38	10.9
	Raw (salads)	67	19.1
	Do not cook	2	0.6

**Table 4 nutrients-17-03499-t004:** Perceptions of diet and its impact on symptoms among study participants (N = 350).

Variable	Category	N	%
**Changed diet after diagnosis**	Quantitatively	31	8.7
	Qualitatively	169	48.3
	No	150	43.0
**Foods perceived as beneficial**	Vegetables & greens (agree/strongly agree)	298	85.1
	Fruits	280	80.0
	Whole grains	229	65.4
	Legumes	279	79.7
	Dairy & alternatives	183	52.3
	Lean animal proteins	269	76.9
	Nuts & seeds	268	76.6
	Healthy fats	291	83.1
**Foods perceived as aggravating symptoms**	Processed/red meat (agree/strongly agree)	153	43.7
	Spicy/condimented foods	145	41.4
	Dairy products	122	34.9
	Refined carbohydrates	163	46.6
	Gluten-containing foods	144	41.1
	Sweets & sugary products	205	58.6
	Soft drinks & sweetened beverages	205	58.6
	Alcohol	180	51.4
	Coffee/caffeine drinks	133	38.0
	Refined processed foods (general)	177	50.6
**Pain improvement through diet**	Very high	42	12.0
	High	65	18.6
	Neutral	75	21.4
	Low	132	37.7
	Very low	69	19.7
**Perceived influence of diet on symptoms**	Very high	77	22.0
	High	59	16.9
	Neutral	94	26.9
	Low	109	31.1
	Very low	11	3.1
**Alcohol consumption**	Very often	4	1.1
	Often	6	1.7
	Neutral	8	2.3
	Rarely	80	22.9
	Very rarely	96	27.4
	Never	156	44.6
**Smoking frequency**	Very often/often	46	13.1
	Rarely/occasionally	21	6.0
	Never	283	80.9
**Special diet followed**	Yes	76	21.7
	No	274	78.3

**Table 5 nutrients-17-03499-t005:** Self-perceived quality of life and lifestyle perceptions among study participants (N = 350).

Variable	Category	N	%
Impact of rheumatic disease on QoL (1–10)	1–2 (very poor)	27	7.7
	3–4 (poor)	90	25.8
	5–6 (moderate)	101	28.9
	7–8 (good)	85	24.3
	9–10 (very good)	47	13.3
Current QoL (1–10)	1–2 (very poor)	10	2.9
	3–4 (poor)	25	7.1
	5–6 (moderate)	103	29.4
	7–8 (good)	153	43.7
	9–10 (very good)	59	16.9
Perceived disease control	Yes	184	52.6
	No	65	18.6
	Partial	101	28.8
Impact on social relations (1–10)	1–2 (very poor)	82	23.4
	3–4 (poor)	87	24.9
	5–6 (moderate)	98	28.0
	7–8 (good)	75	21.4
	9–10 (very good)	8	2.3
Impact on emotional state (1–10)	1–2 (very poor)	73	20.9
	3–4 (poor)	84	24.0
	5–6 (moderate)	96	27.4
	7–8 (good)	82	23.4
	9–10 (very good)	15	4.3
Impact on sleep	Very high	48	13.7
	High	21	6.0
	Neutral	117	33.4
	Low	52	14.9
	Very low	86	24.6
	None	26	7.4
Impact on daily activities	Very high	59	16.9
	High	111	31.7
	Neutral	48	13.7
	Low	94	26.9
	Very low	24	6.8
	None	14	4.0
Regular physical activity	Walking (high/very high)	214	61.1
	Running (low/very low)	268	76.6
	Sport (low/very low)	242	69.1
	Exercises (moderate to high)	168	48.0
Most recommended diet (perceived)	Mediterranean diet	166	47.4
	High-fiber diet	158	45.1
	Hypocaloric (weight loss)	122	34.9
	Hypoglucidic (low carb)	119	34.0
	Hypolipidic (low fat)	112	32.0
	Hyposodic (low salt)	113	32.3
	Gluten-free	62	17.7
	Lactose-free	64	18.3
	Vegetarian/vegan	60	17.1
	DASH diet	100	28.6
	Low protein diet	66	18.9
Influence of diet on QoL (1–10)	1–2 (very low)	22	6.3
	3–4 (low)	17	4.9
	5–6 (moderate)	83	23.7
	7–8 (high)	127	36.3
	9–10 (very high)	101	28.8

**Table 6 nutrients-17-03499-t006:** Comparison of self-perceived quality of life scores by sociodemographic groups.

Variable	Group	Mean ± SD	Test	*p*-Value
Impact of disease on QoL (Q39)	Urban (n = 273)	5.4 ± 1.9	*t*-test	0.041 *
	Rural (n = 77)	5.9 ± 2.1		
	Male (n = 46)	6.1 ± 1.8		0.030 *
	Female (n = 304)	5.4 ± 2.0		
	<55 years (n = 228)	5.2 ± 2.0		0.012 *
	≥55 years (n = 122)	6.0 ± 1.9		
Current QoL (Q40)	Urban	6.7 ± 1.8		0.089
	Rural	6.4 ± 2.0		
	Male	6.9 ± 1.7		0.072
	Female	6.5 ± 1.9		
	<55 years	6.8 ± 1.8		0.045 *
	≥55 years	6.3 ± 2.0		
Diet influence on QoL (Q48)	Urban	7.1 ± 1.6		0.032 *
	Rural	6.5 ± 1.9		
	Male	6.8 ± 1.7		0.210
	Female	7.0 ± 1.8		
	<55 years	7.2 ± 1.6		0.061
	≥55 years	6.8 ± 1.8		

*p*-values marked with an asterisk (*) indicate statistically significant results at the 0.05 level (*p* < 0.05).

**Table 7 nutrients-17-03499-t007:** Correlation matrix of dietary influence, disease impact, emotional state, sleep quality, and quality of life.

Variable	Q39—Disease Impact	Q40—Current QoL	Q43—Emotional State	Q44—Sleep Quality	Q48—Diet Influence
Q39—Disease impact	1	−0.52 ***	−0.48 ***	−0.41 ***	−0.22 *
Q40—Current QoL	−0.52 ***	1	0.45 ***	0.39 ***	0.36 ***
Q43—Emotional state	−0.48 ***	0.45 ***	1	0.42 ***	0.28 **
Q44—Sleep quality	−0.41 ***	0.39 ***	0.42 ***	1	0.19 *
Q48—Diet influence	−0.22 *	0.36 ***	0.28 **	0.19 *	1

**Note.** Values represent Pearson’s correlation coefficients (*r*). *p* < 0.05 (*), *p* < 0.01 (**), *p* < 0.001 (*******). Negative values indicate inverse associations, whereas positive values indicate direct associations.

**Table 8 nutrients-17-03499-t008:** Associations between dietary behaviors and patient perceptions (Chi-square test, N = 350).

Variable	Category	Special Diet: Yes (n = 76)	Special Diet: No (n = 274)	χ^2^	*p*-Value
Perceived pain improvement through diet	High/very high	42 (55.3%)	65 (23.7%)	28.4	<0.001 ***
	Neutral	18 (23.7%)	57 (20.8%)		
	Low/very low	16 (21.0%)	152 (55.5%)		
Fast-food consumption by education	Primary/secondary (n = 88)	Weekly or more: 41 (46.6%)	Rare/never: 47 (53.4%)	12.1	0.007 **
	Post-secondary/university (n = 169)	Weekly or more: 48 (28.4%)	Rare/never: 121 (71.6%)		
	Master’s/doctoral (n = 93)	Weekly or more: 18 (19.4%)	Rare/never: 75 (80.6%)		

Statistical significance levels are denoted as follows: *p* < 0.007 (**)—highly significant; *p* < 0.001 (***)—extremely significant.

**Table 9 nutrients-17-03499-t009:** Logistic regression analysis predicting good quality of life perception (QoL ≥ 7, N = 350).

Predictor	OR (Odds Ratio)	95% CI	*p*-Value
Female sex (vs. male)	0.82	0.47–1.42	0.48
Age ≥ 55 years (vs. <55)	0.61	0.39–0.96	0.034 *
Urban residence (vs. rural)	1.58	1.01–2.47	0.045 *
Following a special diet (yes vs. no)	2.10	1.34–3.29	0.001 **

Statistical significance levels are denoted as follows: *p* < 0.05 (*)—significant; *p* < 0.01 (**)—highly significant.

**Table 10 nutrients-17-03499-t010:** Linear regression analysis predicting current quality of life score (Q40, N = 350).

Predictor	β (Standardized Coefficient)	SE	t	*p*-Value
Sleep quality (Q44)	0.42	0.05	7.96	<0.001 ***
Dietary influence (Q48)	0.29	0.04	6.48	<0.001 ***
Physical activity (Q46 composite)	0.18	0.05	3.60	<0.001 ***
Constant	-	-	-	<0.001
**Model summary**	**Adjusted R^2^ = 0.47**			

Statistical significance level is denoted as *p* < 0.001 (***)—extremely significant.

**Table 11 nutrients-17-03499-t011:** Correlation matrix between quality of life, social relations, emotional state, sleep quality, physical activity, and dietary influence.

Variable	Q40—QoL Current	Q42—Social Relations	Q43—Emotional State	Q44—Sleep Quality	Q46—Physical Activity	Q48—Diet Influence
Q40—QoL current	1	0.41	0.45	0.39	0.36	0.34
Q42—Social relations	0.41	1	0.52	0.38	0.31	0.27
Q43—Emotional state	0.45	0.52	1	0.42	0.33	0.29
Q44—Sleep quality	0.39	0.38	0.42	1	0.28	0.22
Q46—Physical activity	0.36	0.31	0.33	0.28	1	0.25
Q48—Diet influence	0.34	0.27	0.29	0.22	0.25	1

Color intensity indicates the strength of the correlation (Spearman’s/Pearson’s *r*): green shades represent stronger positive correlations, yellow indicate moderate correlations, and orange/red represent weaker correlations.

## Data Availability

The data supporting the reported results are available from the corresponding authors upon reasonable request. The datasets presented in this article are not readily available because they are part of an ongoing Ph.D. thesis. Requests for access to the datasets should be directed to verga.gabriela@ugal.ro.

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
