# Peer review of "Diet, Lifestyle Factors, and Quality of Life in Patients with Rheumatic Diseases: A Cross-Sectional Study"

_nutrients, 2025, doi:10.3390/nu17223499_

Round 1
Reviewer 1 Report
Comments and Suggestions for Authors
Verga GIR et al have undertaken an interesting study on the influence of diet/lifestyle on the quality of life in patients with rheumatic disease.
This manuscript is very well written and contains broad analyses of diet, exercise, sleep etc in a cohort that is well-characterised with regard to age, education and rural/urban locations.
Overall this is a well-conducted study of rheumatic diseases in a patient population that can be segmented by several factors. As mentioned by the authors, there are ample opportunities to take this research forward and to immediately effect interventions that may impact on clinical care, where there is unmet clinical need. An example of this is the difference between rural and urban populations in the quality of life.
Suggestions for consideration.
1
Figure 1 as presented merely demonstrates the distributions of three measures related to quality of life.
This figure doesn't really say a great deal. If the authors wish to keep it, it is suggested that the Y-axis is made ~4x taller and the X-axis compressed by half to better emphasize the height of the distribution.
It is not necessary to have the word "Group" repeated 10 times when it can be used once as an axis title.
2
In Table 8, much is made of the relationship between fast food consumption and educational attainment. This is something of a tangent from the main purpose of the study. Perhaps the authors could further elaborate, in a few sentences, the relevance of this finding as well as any relationship between location (urban/rural) vs educational attainment or some other factor that might explain differences in the quality of life between these populations.
3
As an observational study of a patient population, the authors have clearly outlined some limitations in the Discussion (e.g. causation, bias, "mid-range" sample size etc).
Could the authors also consider elaborating on the lack of comparisons with healthy normal participants? The authors have not specifically mentioned accounting for multiple statistical tests.
4
Minor typographic error on Ln 527 > " ...lifestyle behavioursparticularly nutrition...".
5
The title of Diagram 2 appears to be missing text at the end. Please amend.
6
Question 39 (Ln 869) on the questionnaire is worded awkwardly.
39.
On a scale from 1 to 10, to what extent does your rheumatic condition affect your quality of life?
(1 = very poor, 10 = very good) < this doesn't match the question
Reviewer 2 Report
Comments and Suggestions for Authors
- Diagram2 is not relevant to the result of this study. It should be removed.
- Where did these patients enrolled? It could cause some bias of this study. The authors should mentioned it in the method.
- Patients with rheumatic disease are : Rheumatoid arthritis, Ankylosing spondylitis, Systemic lupus erythematosus, Sjögren’s syndrome, Vasculitis, Osteoarthritis, Gout
Polymyalgia rheumatica, and Scleroderma. It is very heterogeneous. Alochol obviously related to the gout attack. Patients with RA had very high inflammation burden and patients with OA did not. I think this is a big problem for this study. At least tried to limited the diseases number enrolled. Or analysis the difference bewteen the Qol of difference diseases. - Perception of Diet: I think there would be many bias just according to patient's believe and memory.
- What does "Special diet" means?
- This is a cross sectional study. Few conclusions could be obtain from this study.
- In table 9, the author compared the "Use of DMARDs/biologics (yes vs. no)" for the quality of life. It should be removed. Since patients with OA did not have any DMARDs/biologics could be used. This item is inappropriate.
Round 2
Reviewer 2 Report
Comments and Suggestions for Authors This study is acceptable.